# Comprehensive Profiling of Early Neoplastic Gastric Microenvironment Modifications and Biodynamics in Impaired BMP-Signaling FoxL1^+^-Telocytes

**DOI:** 10.3390/biomedicines11010019

**Published:** 2022-12-22

**Authors:** Alain B. Alfonso, Véronique Pomerleau, Vilcy Reyes Nicolás, Jennifer Raisch, Carla-Marie Jurkovic, François-Michel Boisvert, Nathalie Perreault

**Affiliations:** Département d’Immunologie et Biologie Cellulaire, Faculté de Médecine et des Sciences de la Santé, Université de Sherbrooke, Sherbrooke, QC J1E 4K8, Canada

**Keywords:** FoxL1^+^-telocytes, epithelial–mesenchymal interaction, BMP signaling, extracellular matrix, mechanical microenvironment, matrisome

## Abstract

FoxL1^+^telocytes (TC^FoxL1+^) are novel gastrointestinal subepithelial cells that form a communication axis between the mesenchyme and epithelium. TC^FoxL1+^ are strategically positioned to be key contributors to the microenvironment through production and secretion of growth factors and extracellular matrix (ECM) proteins. In recent years, the alteration of the bone morphogenetic protein (BMP) signaling in TC^FoxL1+^ was demonstrated to trigger a toxic microenvironment with ECM remodeling that leads to the development of pre-neoplastic gastric lesions. However, a comprehensive analysis of variations in the ECM composition and its associated proteins in gastric neoplasia linked to TC^FoxL1+^ dysregulation has never been performed. This study provides a better understanding of how TC^FoxL1+^ defective BMP signaling participates in the gastric pre-neoplastic microenvironment. Using a proteomic approach, we determined the changes in the complete matrisome of *BmpR1a*^△FoxL1+^ and control mice, both in total antrum as well as in isolated mesenchyme-enriched antrum fractions. Comparative proteomic analysis revealed that the deconstruction of the gastric antrum led to a more comprehensive analysis of the ECM fraction of gastric tissues microenvironment. These results show that TC^FoxL1+^ are key members of the mesenchymal cell population and actively participate in the establishment of the matrisomic fraction of the microenvironment, thus influencing epithelial cell behavior.

## 1. Introduction

The extracellular matrix (ECM) is a complex assembly of large fibrous proteins, glycoproteins, proteoglycans, and ECM-associated proteins, such as growth factors, whose composition varies from one tissue to another [1]. The ECM represents the insoluble fraction of the microenvironment, and although it was long believed to be a passive component, it is in fact highly dynamic and influences the behavior of neighboring cells through mechanosensing and signaling [2,3]. Thus, the architecture and homeostasis of a tissue, such as the stomach, are maintained in part by tight regulation of ECM dynamics. Dysregulation of the ECM composition in the microenvironment creates a disbalance in the physical (force, porosity, stiffness) and biochemical (growth factor density, cell adhesion, signaling) stimuli, providing an abnormal cell response to these biomechanical forces and leading to the development of diseases such as gastric neoplasia [4,5,6,7,8]. In gastric cancer, pre-malignant lesions already show dysregulation in ECM dynamics and will also influence the prognosis outcome and therapeutic strategies at later stages of the disease [2,5,9].

In mammals, the ECM is composed of approximately 300 proteins. This represents the core matrisome, which is mainly composed of proteins, such as collagens (CLs) and proteoglycans, with structural and fibrillar glycoproteins [10,11,12,13]. The biochemical properties of these proteins, such as their size, insolubility, and cross-linking, have made attempts to systematically characterize the entire tissue ECM composition challenging [14]. Recently, Naba et al. developed a proteomics-based approach to identify, quantify, and compare the matrisome of whole tissues, partially resolving the limitations of in vivo analysis of ECM dynamics [14]. This approach allows for comprehensive evaluation of the proteins from the core matrisome, as well as the components of matrisome-associated proteins such as ECM regulators (ECM-remodeling enzymes, cross-linkers, proteases) and secreted factors such as growth factors and cytokines binding the ECM [13,14].

As the microenvironment plays an essential role in tissue homeostasis and in the development of pathologies such as gastric cancer [4,5,6,7,8], mesenchymal cells have attracted considerable attention in recent years [15,16,17]. Mesenchymal cells, more precisely myofibroblasts as well as FoxL1^+^telocytes (TC^FoxL1+^), are better known for their contribution to the sub-epithelial microenvironment. Both myofibroblasts and TC^FoxL1+^ are capable secretors of cytokines, chemokines, growth factors, and ECM proteins [16,17,18,19,20,21,22]. In addition, TC^FoxL1+^ are advantageously positioned directly underlying the epithelium, forming a 3D nexus between the epithelium and the rest of the stroma [17,23]. TC^FoxL1+^ contribute to the stem cell niche microenvironment by secreting soluble factors such as WNT5a, R-spondin3, and gremlin, which has been documented in recent years [15,17,20,23,24]. However, the precise role of TC^FoxL1+^ in the insoluble fraction of the gastrointestinal (GI) microenvironment is poorly defined. Considering the effect of TC^FoxL1+^ on GI epithelial cells [17,18,19,22,23], there is a critical need to rigorously characterize the role of the ECM biodynamic microenvironment on GI epithelial cell behavior in vivo and determine the contribution of TC^FoxL1+^.

To date, there have been limitations to the study of the various roles of TC^FoxL1+^ in the in vivo microenvironment because of the limited models available [17,20,23,25,26]. A previous study, using a murine model with TC^FoxL1+^ impaired BMP signaling pathway, demonstrated the importance of these cells and this pathway in inducing gastric neoplastic lesions and polyps in 90-day-old mice [22]. *BmpR1a*^△FoxL1+^ mice did not develop chronic inflammation or a malignant phenotype; however, disturbed TC^FoxL1+^ led to early precancerous events with important disorganized gastric glands architecture, intestinal metaplasia, and spasmolytic polypeptide-expressing metaplasia (SPEM), in addition to remodeling of the ECM into a reactive microenvironment [22]. Consequently, *BmpR1a*^△FoxL1+^ mice represent an excellent model to investigate the TC^FoxL1+^ contribution instructing the microenvironment ECM biodynamics, leading to gastric neoplasia. Using this model, we can perform a matrisomic investigative of the stomach of control and *BmpR1a*^△FoxL1+^ mice, and better understand the contribution of TC^FoxL1+^ to this aspect of the microenvironment [13,14].

In the present study, we evaluated the contribution of TC^FoxL1+^ to the matrisomic microenvironment in mice with early gastric neoplasia. This matrisomic investigative approach, used in concert with the TC^FoxL1+^ signaling impaired gastric pre-neoplastic mouse model, revealed a detailed inventory of dysregulated core-matrisome and matrisome-associated proteins in early events of gastric neoplasia. We identified important and subtle changes in the ECM biology that occur during the etiology of gastric neoplasia associated with Bmp-signaling impaired TC^FoxL1+^.

## 2. Materials and Methods

### 2.1. Animals

The transgenic mouse line C57BL/6J *FoxL1*Cre was provided by Dr. Kaestner [27] and 129 SvEv-*BmpR1*afx/fx mice were supplied by Dr. Mishina [28]. *BmpR1a*^ΔFoxL1+^ conditional knockout mice were generated as previously described [18,21,22]. Male and female 90-day-old age-matched mice were used for the study. All experiments were performed in accordance with our animal welfare protocol (approval number: FMSS-2019-2370).

### 2.2. Deconstruction of Mouse Ex Vivo Stomach Tissues

Tissue deconstruction was performed stepwise to enrich each compartment (the epithelial, mesenchymal, and muscular layers). First, stomachs were opened along the greater curvature and rinsed with cold 1× PBS, and the antrums were isolated from the corpus and fundus sections of the total tissue. Mouse antrums were cut with a razor blade into 5 mm tissue sections and the muscle layer was mechanically dissociated using forceps under a stereomicroscope. Leftover tissues (mesenchyme and epithelium) were subsequently incubated in 4 mL sterile Corning^TM^ Cell Recovery Solution without agitation (Corning Life Science, Corning, NY, USA) at 4 °C for 24 h. The following day, dissociation of the epithelial layer was performed with a 30 min incubation of the tissue on ice followed by vigorous manual shaking for 15 s. The mesenchymal tissue was incubated once again in 6 mL of sterile Corning^TM^ Cell Recovery Solution (Corning Life Science, Corning, NY, USA) on ice with gentle shaking for 30 min followed by further dissociation by vigorous manual shaking for 15 s. Finally, mesenchymal tissues were washed four times with 1× PBS while all remaining epithelial cells were pooled and kept on ice. Deconstructed tissue sections were either snap-frozen for immunoblotting and proteomic analysis or fixed in 4% paraformaldehyde (PFA) (Thermo Fisher Scientific, Waltham, MA, USA) and paraffin-embedded for histological analysis. Total tissue samples were also collected to allow for a more comprehensive comparison of the matrisome content.

### 2.3. Histological Analysis

The total stomach antrum or deconstructed fractions were fixed overnight at 4 °C in 4% PFA (Thermo Fisher Scientific, Waltham, MA, USA) and subsequently processed for tissue embedding as previously described [18,21]. To avoid the diffusion of cells in paraffin, the epithelial layer from the deconstructed tissue was embedded in HistoGelTM (Thermo Fisher Scientific, Waltham, MA, USA) and wrapped in lens paper prior to embedding. Histological staining (H&E) on tissue sections was performed as previously described [18,21]. Virtual images were acquired with a slide scanner (Nanozoomer; Hamamatsu, Japan) and visualized using the NDP.view2 software (version 2.8.24).

### 2.4. In-Solution Digestion of Proteins to Peptides for Mass Spectrometry Analysis

Frozen samples of either the total stomach antrum or mesenchymal-enriched stomach antrum fractions were thawed on ice and homogenized directly in 8 M urea (Sigma Aldrich, St. Louis, MO, USA) dissolved in 10 mM HEPES pH 8.0 (Wisent, Saint-Jean-Baptiste, QC, Canada) (100 µL/10 mg wet tissue weight), using the QIAGEN TissueLyser LT (Hilden, Germany). Prior to protein quantification by BCA assay (Pierce Thermo Scientific, Waltham, MA, USA), samples were centrifuged following their homogenization to remove urea-insoluble materials. Following the protocol described by Naba et al., proteins were reduced, alkylated, deglycosylated, and digested, except for the Lys-C digestion, which was omitted [14,29]. Solutions were prepared using MS-grade water and low protein binding tubes were used for these experiments.

### 2.5. Purification and Desalting of the Peptides on C18 Columns

Trifluoroacetic acid (TFA) was added following incubation with the proteases to a final concentration of 0.2%, and the samples were desalted using C18 tips (Pierce Thermo Scientific, Waltham, MA, USA). Acetonitrile was first aspirated in the C18 tip initially and then equilibrated with 0.1% TFA. Each peptide sample was bound to the C18 tip by 10 successive up-and-down until the entire sample was loaded. The tip was then washed with a solution containing 0.1% TFA, and the peptides were eluted in a separate low-bind tube using a 50% acetonitrile/1% formic acid solution. The eluted peptides were lyophilized using a centrifugal evaporator at 60 °C and the dry peptides were resuspended in 1% formic acid. The peptide concentration was measured using a NanoDrop spectrophotometer (Thermo Fisher Scientific, Waltham, MA, USA) at 205 nm absorbance. The peptide samples were transferred to autosampler vials and stored at −20 °C until analyzed by mass spectrometry.

### 2.6. LC-MS/MS Analysis

Analysis of the purified peptides was carried out at the Université de Sherbrooke proteomics facility using the following parameters: Each sample (was injected into an HPLC system (NanoElute, Bruker Daltonics, Billerica, MA, USA) for LC-MS/MS. A total of 250 ng of peptides were loaded onto a trap column at a constant flow of 4 µL/min (Acclaim PepMap100 C18 column, 0.3 mm id × 5 mm, Dionex Corporation, Sunnyvale, CA, USA) and eluted onto the C18 analytical column (1.9 µm beads size, 75 µm × 25 cm, PepSep) over a 2 h gradient of acetonitrile (5–37%) in 0.1% FA at 500 nL/min into a TimsTOF Pro ion mobility mass spectrometer equipped with a Captive Spray nanoelectrospray source (NanoElute, Bruker Daltonics, Billerica, MA, USA). The data were acquired in data-dependent MS/MS mode with a 100–1700 m/z mass range, and the number of PASEF scans was set at 10 (1.27 s duty cycle) with a dynamic exclusion m/z isolation window of 0.4 min. The collision energy was set at 42.0 eV, and the target intensity was 20,000 with an intensity threshold of 2500.

### 2.7. Protein Identification Using MaxQuant Analysis

MaxQuant software version 1.6.17 (Munich, Bavaria, Germany), was used to analyze the raw files using the Uniprot mouse proteome database (25 March 2020, 55,366 entries). The analysis was performed under TIMS-DDA type in group-specific parameters, and included the following parameters: two miscleavages were allowed; fixed modification was carbamidomethylation of cysteine; the enzyme selected was trypsin (not before a proline). The following variable modifications were included in the analysis: methionine oxidation, N-terminal protein acetylation, and protein carbamylation (K, N-terminal). The limit for mass tolerance was set at 10 ppm for the precursor ions and at 20 ppm for the fragment ions. The identification values “PSM FDR”, “Protein FDR”, and “Site decoy fraction” were set to 0.05. The minimum peptide count was set to 1. Label-free quantification (LFQ) was performed using an LFQ minimal ratio count of 2. Both the “Second peptides” and “Match between runs” were allowed.

### 2.8. Differential and Statistical Analyses of Mass Spectrometry Data

Following the MaxQuant analysis, LFQ intensities were sorted according to several parameters using the Prostar software version 1.28.1 (Grenoble, France) [30]. Filtered proteins positive for the “Reverse”, “Only.identified.by.site”, or “Potential.contaminant” categories were eliminated, as were proteins identified from only one unique peptide. Data were normalized with quantile centering set to 0.5 for the intensity distribution. The non-detection of a protein was considered biologically relevant in the following cases: 75% (3 of 4) of the control or mutant mice group with respect to the other for total antrum (TA) and in 83% (5 of 6) of the control or mutant mice group with respect to the other for enriched mesenchyme (EM). Considering the aforementioned conditions, for all data corresponding to the matrisome, the partially observed value (POV) imputation was revised according to the following cases, followed by recalculation of Log2FC and *p*-value in ProStar. For TA data, the imputed POV was removed and replaced by the minimum POV when three out of four mice presented an LFQ intensity = 0 for a given protein. If two out of four mice presented LFQ intensity = 0, the Log2FC and the *p*-value recalculated in ProStar were considered non-conclusive (NC). For the EM data, the imputed POV was removed and replaced by the minimum POV when five out of six mice in one of the two groups presented an LFQ intensity = 0 for a given protein. If four out of six mice presented an LFQ intensity = 0 in one or both groups, the Log2FC and *p*-value recalculated by ProStar were considered NC. Structured least square adaptation (SLSA) and detQuantile imputation were performed for POV and missing values in the entire condition (MEC), respectively. The results were ranked to preserve the proteins present in at least three of the four (in TA) and three of the six (in MS EM) biological replicates for each condition. For hypotheses testing, a Limma statistical test was used, with a fold-change threshold of 1.5 and a *p*-value of 0.05, to determine the list of differentially abundant proteins. A “st.boot” calibration plot was chosen for *p*-value distribution.

### 2.9. Matrisome Identification

The Matrisome Annotator webtool (matrisomeproject.mit.edu) was used to annotate the list of differentially abundant proteins as previously described [13]. Matrisome divisions (core matrisome or matrisome-associated) and categories (ECM glycoproteins, collagens (CLs) and proteoglycans, ECM-affiliated, ECM regulators, and secreted factors) were used according to Naba et al. [13].

### 2.10. Indirect Immunofluorescence

Indirect immunofluorescence of stomach sections from 90-day old control and *BmpR1a*^△FoxL1+^ mice was performed as previously described [18,21,22,31,32,33]. Antigen blocking was performed with a solution of 2% bovine serum albumin (BSA), 0.1% fish gelatin, and 0.2% Triton X-100 in 1× PBS for 1 h at room temperature. The following primary antibodies were used in this study: S100A9 (Cell Signaling Technology, Danvers, MA, USA; Cat#73425; RRID:AB_2799839), fibronectin (Millipore; Burlington, MA, USA, Cat# AB2033, RRID:AB_2105702), and tenascin C (Millipore, Burlington, MA, USA; Cat# AB19013, RRID:AB_2256033). The following day, slides were incubated with anti-rabbit IgG Alexa-488 labeled secondary antibody (Cell Signaling Technology; Danvers, MA, USA; Cat# 4412; RRID:AB_1904025). Slides were examined under a Zeiss Axioscope 5 (Oberkochen, Germany) equipped with a Zeiss Axiocam 705 mono CMOS camera. Images were analyzed using ImageJ v.1.53j (RRID:SDR_003070).

### 2.11. Picro-Sirius Red Staining

Tissue sections of 90-day old mouse stomach were stained with picrosirius red following a previously published protocol [34] and CL content and fibers were analyzed under bright-field and polarized light. Images from four mice in each group were taken using a Zeiss Axioscope 5 equipped with a linear polarizer and analyzer. Multiple representative regions of interest (ROI) were assessed per image to characterize the alignment properties of CL fibers. ROI were selected in both the top and middle antrum glands of *BmpR1a*^△FoxL1+^ mice to better assess tissue complexity. Each ROI was the same dimension. The distribution of CL fiber angles and coherency was determined using ImageJ software (Madison, WI, USA) package Orientation J (version 2.0.5; RRID:SCR_014796). Statistical analysis was performed using Prism v9.4.1 (San Diego, CA, USA, RRID:SCR_002798). To test the normal distribution of the samples, we used D’Agostino-Pearson omnibus normality test and for group analyses we used nested ANOVA.

### 2.12. Immunoblot Analysis

The same 8 M urea proteins extracts from total antrum tissues used for proteomic analyses were also assessed to validate the potential proteins of interest (*n* = 4). Samples (10 μg each) were separated on NuPage 4–12% Bis-Tris gels (Thermo Fisher Scientific, Waltham, MA, USA) with MES buffer and transferred onto a PVDF membrane. Membranes were probed with the following antibodies: S100A8 (Proteintech, Rosemont, IL, USA; Cat# 15792-1-AP, RRID:AB_10666315), S100A9 (Cell Signaling Technology; Danvers, MA, USA; Cat#73425; RRID:AB_2799839), SPARCL1 (R&D Systems, Minneapolis, MN, USA; Cat# AF2836, RRID:AB_2195097), and ADAM9 (Cell Signaling Technology; Danvers, MA, USA; Cat# 4151, RRID:AB_1903892). GAPDH (Cell Signaling Technology; Danvers, MA, USA; Cat# 2118, RRID:AB_561053) was used as a loading control. Anti-rabbit (Cat#7074; RRID:AB_2099233) HRP-labeled secondary antibodies were purchased from Cell Signaling Technology; Danvers, MA, USA; and anti-goat HRP-labeled antibodies (Cat#705-035-003; RRID:AB_2340390) were from Jackson ImmunoResearch Laboratories (West Grove, PA, USA). Immunoreactive bands were detected using the Amersham ECL Western blotting Detection System (GE Healthcare Life Sciences/Cytiva, Chicago, IL, USA) with an Azure Biosystems c280 digital imager (Azure Biosystems, Dublin, CA, USA). Quantification was performed using ImageJ v1.53j (*n* = 4 mice/group). The Mann–Whitney U test was used to determine data significance.

## 3. Results

To study the contribution of TC^FoxL1+^ in instructing the microenvironment ECM biodynamic leading to gastric neoplasia through a matrisomic investigative approach, we compared and analyzed two methods for tissue preparation of the stomach antrum of 90-day-old control and *BmpR1a*^△FoxL1+^ mice (Figure 1A). In the first approach, an 8 M urea extraction of total proteins was performed on the stomach antrum of the control and *BmpR1a*^△FoxL1+^ mice. Proteins from the total antrum were identified using LC-MS/MS as previously described [14]. For the second method, we investigated whether other cell compartments in the tissue caused unwanted interference during the protein identification and quantification within the proteomic analysis. As the bulk of ECM/matrisome proteins is located in the mesenchymal compartment, we decided to deconstruct the stomach antrum to obtain an enriched mesenchymal compartment (Figure 1B–E). First, the stomach antrum was isolated from control and *BmpR1a*^△FoxL1+^ mice (Figure 1B), and the muscle layers (Figure 1C) were mechanically separated from the antrum using tweezers. Next, the remaining epithelium/mesenchymal fraction (Figure 1D) was incubated with a non-enzymatic cell recovery solution that dissociated the epithelial fraction (Figure 1E) from the underlying mesenchyme, as previously described [18,32,33,35]. The 8 M protein extraction was carried out for the isolated enriched mesenchymal fraction, and the analysis was performed as described above for the total tissue.

### 3.1. Analysis of the Matrisome from Total Antrum of BmpR1a^△FoxL1+^ Mouse

To evaluate the changes in ECM composition in our pre-neoplastic gastric *BmpR1a*^△FoxL1+^ mouse model, we calculated the fold change in matrisome proteins between the total antrum of mutant and control mice. The ratio (*BmpR1a*^△FoxL1+^/control) of relative expression of total proteins between both groups was compared. Among the 3803 proteins detected, 279 were shown to be upregulated, while 484 were downregulated (Figure 2A). The analysis identified, from the total antrum, the presence of 36 overexpressed matrisome proteins (dark red spots, FC > 1.5) and 37 downregulated proteins (dark blue spots, FC < −1.5) in *BmpR1a*^△FoxL1+^ mice compared to those observed in the control group (Figure 2A). Matrisome proteins were identified using the Matrisome Annotator analytical tool (http://matrisomeproject.mit.edu/; accessed on 29 September 2020) [13,14,36]. A total of 169 proteins were identified, 70 of them belonging to the core matrisome and 99 to matrisome-associated proteins. Of the proteins belonging to the core matrisome, we identified 11 proteoglycans, 10 CLs, and 49 glycoproteins, whereas we identified 28 ECM-affiliated proteins, 54 ECM regulators, and 17 secreted factors among the matrisome-associated proteins (Figure 2B). Surprisingly, except for two the CL chains (CL1α2, CL4α1, and α2; CL6α1, α2, and α5; CL12α1 and CL14α1) in *BmpR1a*^△FoxL1+^ mice, all were downregulated compared to those observed in controls (Table 1). Only CL15α1 and CL18α1 were upregulated in the mutant mice compared to those in the controls (Table 1). Similarly, most proteoglycans (HSPG2, perlecan; ASPN, asporin; DCN, decorin; LUM, lumican; and VCAN, versican) were observed to be negatively modulated in *BmpR1a*^△FoxL1+^ mice compared to those in the controls. Only biglycan (BGN) and bone marrow proteoglycan (PRG2) were upregulated in the mutant mice compared to those in the controls (Table 1) Glycoproteins such as Agrin (AGRN), fibronectin I (FNI), tenascin C (TNC), vitronectin (VTN), and periostin (POSTN) were upregulated in mutant mice compared to those in the controls, whereas others such as microfibrillar-associated proteins (MFAP2, 4, and 5), Nidogen1 and 2 (NID1 and NID2), as well as SPARC-like protein-1 (SPARCL-1) were downregulated (Table 1). Among the matrisome-associated proteins, the analysis revealed that ECM-affiliated proteins such as proteins of the annexin family including annexin 10 (ANXA10) and different galectins, such as galectin-4 (LGALS4) and mucin 4 (MUC4), were upregulated, whereas annexin 6 (ANXA6) and chondroitin sulfate proteoglycan 4 (CSPG4) were downregulated in *BmpR1a*^△FoxL1+^ mice compared to those in the controls (Table 1). Analysis of ECM regulators revealed that disintegrin, metalloproteinase family members (ADAM9 and 10), and various serpins (SERPINB1a, SERPINB5, and SERPINB12) were overexpressed, whereas α-1-microglobulin/bikunin (AMBP) and transglutaminase 2 (TGM2) were downregulated in mutant mice compared to those in the controls (Table 1). For the secreted factors, proteomic analyses showed that most members of the S100 protein group (S100A1, A2, A4, A6, A8, A9, A11, A13, A14, A16, and G) were overexpressed, except for S100B, which was downregulated in *BmpR1a*^△FoxL1+^ mice compared to that measured in controls (Table 1).

### 3.2. Analysis of the Matrisome from Enriched Mesenchymal Antrum of BmpR1a^△FoxL1+^ Mouse

Next, we evaluated changes in the ECM composition of antrum-enriched mesenchyme extracts from both mutant and control mice. We detected 37.5% fewer proteins in the enriched mesenchyme (2377) compared to those in the total antrum (3803); however, we discovered that a greater number of proteins were modulated, with 827 being upregulated and 492 being downregulated (Figure 3A). The analysis of the enriched mesenchymal antrum revealed the presence of 34 overexpressed matrisome proteins (dark red spots, FC > 1.5) and 59 downregulated proteins (dark blue spots, FC < −1.5) in *BmpR1a*^△FoxL1+^ mice compared to those in the control group (Figure 3A). As described above, matrisome proteins were identified using the Matrisome Annotator analytical tool (access date: 15 December 2020). A total of 135 proteins were identified, of which 68 belonged to the core matrisome and 67 to the matrisome-associated proteins. Of the proteins belonging to the core matrisome, we identified 10 proteoglycans, 12 CLs, and 46 glycoproteins, whereas among the matrisome-associated proteins, 21 ECM-affiliated proteins, 34 ECM regulators, and 12 secreted factors were identified (Table 2). As observed for the total tissue extract, most CL chains (CL1α1, CL4α1, CL6α1, α2, α3 and α5, and CL15α1) and most proteoglycans (perlecan, asporin, decorin, lumican, and versican) in the antrum enriched mesenchyme were downregulated in *BmpR1a*^△FoxL1+^ mice compared to those in the controls (Table 2). We observed that, unlike the total antrum extract, biglycan was downregulated in the enriched mesenchymal antrum extract from mutant mice compared to that from controls (Table 2). Similar results were obtained with the enriched mesenchymal antrum extract for glycoproteins. FN1, TNC, and VTN were upregulated, whereas MFAP2, 4, and 5, NID1 and NID2, and SPARCL-1 were downregulated in mutant mice compared to those measured in controls (Table 2). However, in the enriched mesenchymal antrum extract, Agrin was downregulated, in contrast to our observations for the total antrum extract. Finally, our analysis of the matrisome-associated proteins, ECM-affiliated proteins, ECM regulators, and secreted factors revealed variations in mostly similar proteins identified in the total tissue extract (Table 2). When we compared both analyses, we discovered that the matrisomic variations obtained from the enriched mesenchymal antrum extracts were more robust than those obtained from the total antrum extract.

Data from both types of tissue extracts analyzed were further processed to remove irrelevant data, which led to the identification of 184 matrisome proteins between both experiments (Figure 4). Venn diagrams of the different protein categories, core matrisome (in green), and matrisome-associated proteins (in black), revealed that mesenchymal enrichment did not lead to heavy loss of matrisomal proteins in relation to the total tissue extract, except for the ECM regulators, which were more affected by the tissue treatment. Next, we performed a functional association network using the STRING database and the 116 matrisome proteins that were identified to be significantly modulated in both experiments to obtain a signature profile of proteins indicative of biological processes occurring in the microenvironment of our mouse model. The STRING analysis revealed changes in proteins involved in immune regulation, fibrosis, and tumor microenvironment in *BmpR1a*^△FoxL1+^ mice compared to those in controls (data not shown).

### 3.3. Loss of BMP Signaling in Gastric TC^FoxL1+^ Induces Dysregulations in ECM Biodynamics Associated with Inflammation

The tissue microenvironment can play an important role in cellular behavior, and ECM proteins influence the biodynamics as well as cell biology of tissues [37,38,39]. The core matrisome proteins’ influence on the microenvironment through biomechanical and biochemical sensing is evident. However, it is important to take into consideration that the ECM can act as a reservoir for secreted growth factors, chemokines, and cytokines also affecting the microenvironment and impacting cell behavior [37,39]. Histopathologically, *BmpR1a*^△FoxL1+^ mice have been shown to be more prone to gastric neoplasia with mild inflammation [22]. Here, a part of the functional network analysis suggested a protein signature profile linked to immune regulation. S100A8 and S100A9, both secreted factors associated with the ECM, have been associated with acute and chronic inflammatory conditions and autoimmune diseases [40,41,42]. Matrisomic profiling revealed a significant increase in S100A8 and S100A9 between *BmpR1a*^△FoxL1+^ mice and controls in total antrum (FC = 11,412 and 13058, respectively; Table 1) as well as in the enriched mesenchymal antrum (FC = 37.9 and 85.2, respectively; Table 2). S100A9 expression in mutant mice was confirmed through immunofluorescence, with strong expression in the *BmpR1a*^△FoxL1+^ mouse mesenchyme, whereas controls showed no expression of the protein (Figure 5A). In addition, immunoblot analysis against secreted factors S100A8 and A9 revealed de novo expression of both proteins in the mutant mice but not controls, where these proteins were not detected (fold change = 20.34 and 20.48, respectively; Figure 5B,C).

### 3.4. Disruption of the CL Network in Mice with Impaired Gastric BMP Signaling in TC^FoxL1+^

CL is a dominant and important element in the pathological microenvironment and has a significant influence on the initiation and development of pathologies such as cancer [10]. Furthermore, its expression is generally increased in gastric cancers [43]. However, as shown in Table 1 and Table 2, the expression of almost all CL chains was negatively modulated in *BmpR1a*^△FoxL1+^ mice compared to that in controls (CL1α2, CL4α1, and α2; CL6α1, α2 and α5; CL12α1 and CL14α1). Only a few examples were observed to be positively modulated in mutant mice using both tissue preparation methods (Table 1 and Table 2). These results differ from previously published work with this mouse model [22], in which marked expression and accumulation of CLI and IV in the gastric glands of *BmpR1a*^△FoxL1+^ mice were observed. Therefore, we decided to perform further analyses of the CL network in both mouse groups. Collagen deposition, fiber orientation, and spatial distribution were analyzed using picrosirius red staining under bright and polarized light microscopy in both control and mutant mice (Figure 6). The loss of BMP signaling in TC^FoxL1+^ mice affected the sub-epithelial CL fiber network in mutant mice, mainly towards the upper part of the gland, compared to controls, as shown following picrosirius red staining under bright field (Figure 6A, left panels). Visualization of CL fibers orientation and alignment was performed with polarized light, where fibrillar CL appeared in a range of colors from red, yellow, orange, and green (Figure 6A middle panels). Heterogeneous organization of CL fibers was observed in *BmpR1a*^△FoxL1+^ mice, with areas of increased alignment of fibrillar collagen towards the top of the gland compared to that in controls (Figure 6A middle and right panels). Analysis using the OrientationJ plugin in ImageJ indicates a similar distribution of fiber angles between the control and *BmpR1a*^△FoxL1+^ mice in the middle part of the glands (Figure 6B). However, the upper gland of the mutant mice revealed a divergent spatial organization of CL fibers with respect to the organization observed in the controls (Figure 6C). The coherency factor was significantly higher in the top of the gland in *BmpR1a*^△FoxL1+^ mice (CF = 0.338), indicating that the CL fibers tended to be in a predominant direction and had an increased alignment compared to that observed in control mice (CF = 0.245; Figure 6D).

### 3.5. Loss of BMP Signaling in Gastric TC^FoxL1+^ Causes Remodeling of ECM Glycoproteins Associated with Early Gastric Neoplasia

ECM glycoproteins and ECM regulators are other matrix components essential for proper tissue function, including the stomach [10,44]. In addition, part of the functional annotation analysis also suggested a protein signature profile linked to the tumor microenvironment. Over the years, several ECM glycoproteins and ECM regulators have been associated with every stage of gastric cancer [45,46,47]. Matrisomic profiling revealed a significant increase in ECM glycoproteins such as FN1 between *BmpR1a*^△FoxL1+^ and control enriched mesenchymal antrum (FC = 1.46; Table 2) and TNC in total antrum (FC = 1.4; Table 1) as well as in enriched mesenchymal antrum (FC = 1.95; Table 2). A significant decrease in SPARCL-1 in total antrum (FC = −15377; Table 1) was also observed. Finally, we identified a significant increase in the ECM regulator, ADAM9, only in the in total antrum (FC = 510; Table 1). FN1 (Figure 7A) and TNC (Figure 7B) exhibited increased expressions in *BmpR1a*^△FoxL1+^ mice compared to those in controls, as confirmed by immunofluorescence of stomach sections (Figure 7A,B). The immunoblot analysis against SPARCL-1 confirmed a significant decrease in this ECM glycoprotein in mutant mice compared to that measured in controls (fold change = 0.48; Figure 7C,D). Immunoblot analysis against ADAM9 confirmed a significant increase in this ECM regulator in *BmpR1a*^△FoxL1+^ mice compared to that in controls (fold change = 1.976; Figure 7C,D).

## 4. Discussion

Due to the complexity and extremely low solubility of the ECM, exhaustive biochemical characterization of tissues has long been a challenge. In recent years, mass spectrometry has been used to characterize ECM proteins in various tissues [14,48,49,50]. In addition, the developments brought forward by Naba et al. of an in silico definition of the matrisome provide a possibility for a detailed characterization of the biochemistry and composition of the ECM in normal and diseased tissues [13,14,48,51]. Similar to other diseases, ECM deregulation has been shown to play a role in gastric neoplasia by creating a favorable microenvironment for the transformed cells to thrive from pre-neoplastic lesions to metastatic stages [5,52]. Recent studies have demonstrated that TC^FoxL1+^ are strong contributors to the GI microenvironment [15,17,20,23,24]; however, their precise contribution to the ECM fractions of the microenvironment is less clear. Qualitative analysis of ECM proteins in the *BmpR1a*^△FoxL1+^ mouse, where TC^FoxL1+^ are impaired in BMP signaling, suggests a potential role for this mesenchymal cell population in contributing to the ECM fraction of the microenvironment [18,21,22]. In addition, the pathophysiological phenotype of the *BmpR1a*^△FoxL1+^ mouse model is characterized by the development of gastric pre-neoplastic lesions [22]. Together, we discovered that *BmpR1a*^△FoxL1+^ mice represent an adequate model for understanding how TC^FoxL1+^ participates in an aberrant gastric pre-neoplastic ECM microenvironment.

As part of our study was to characterize the ECM contribution of BMP-signaling impaired TC^FoxL1+^ to the pre-neoplastic gastric microenvironment, we explored the validity of using enriched mesenchyme over total tissue extract for targeting matrisomic proteins. Tissue deconstruction into minimal mesenchymal compartment, where TC^FoxL1+^ and the microenvironment are observable, allows for the possibility of circumventing the complexity of the total tissue protein content. As expected, we observed an important decrease in the presence of ECM regulator proteins when we used enriched mesenchymal extract in comparison to the total tissue extract because these proteins are not bound to the ECM. Thus, they are easily lost during purification processes [48]. Deconstruction of the gastric antrum provides a more comprehensive analysis of the matrisome in *BmpR1a*^△FoxL1+^ mice compared to controls, with the removal of background noise from non-matrisomic proteins. In addition, the mesenchymal-enriched extract allows for improved identification of proteins with low expression levels that could be easily lost in a larger pool of proteins.

In a previous study, the gastric pathophysiological aspects of the *BmpR1a*^△FoxL1+^ mouse model showed that disruption of BMP signaling in TC^FoxL1+^ led to the creation of a toxic microenvironment with an increase in CLI, fibronectin, HGF, and FSP1/S100A4, pressuring the epithelium to initiate pre-malignant lesions [22]. Correa’s cascade of gastric carcinogenesis shows that a normal gastric epithelium gradually transitions from initial gastritis to chronic gastritis, mucosal atrophy, metaplasia, dysplasia, and carcinoma [53,54]. Early steps of this cascade prior to carcinoma involve the presence of inflammatory processes [54,55] and a reorganization of the nurturing microenvironment into a tumor microenvironment [5]. Interestingly, some protein profiles, such as immune regulation, fibrosis, and tumor microenvironment, were noticeably modulated in the *BmpR1a*^△FoxL1+^ matrisome analysis. Thus, the present protein profile, in combination with our previous phenotypic analysis of *BmpR1a*^△FoxL1+^ mice, allows for a better understanding of the sequence of events occurring in the ECM microenvironment of these mice with BMP-impaired TC^FoxL1+^ with regard to early events in gastric neoplasia.

Consequently, the overexpression of S100A8 and A9 in the matrisomic analysis, as secreted factors associated with the ECM, supports these profiles. Both proteins have been associated with numerous human disorders, including acute and chronic inflammatory conditions, autoimmune diseases, and cancer [40,56,57]. They are also reported to represent highly potent biomarkers of a wide range of inflammatory processes, including rheumatoid arthritis and inflammatory bowel disease [41,58]. In tumor biology, both proteins play a fundamental role, and their levels are elevated in numerous tumors, including gastric cancer, which is in line with our model [57,59,60,61,62,63]. Although there are signs of inflammation in mice with infiltration of lymphocytes (CD3) and macrophages (F4/80), no chronic inflammation was observed [22]. This could partially explain the overexpression of S100A8/A9 in the gastric microenvironment of the *BmpR1a*^△FoxL1+^ mice.

As for the tumor microenvironment profile identified in this study, ECM glycoproteins and ECM regulators are known to play key roles in the microenvironment for proper tissue function including the stomach [2,5,10,45,64,65,66]. For example, matricellular proteins such as FN1, TNC, and ADAM9 were upregulated, while SPARCL-1/Hevin was downregulated. In addition, these ECM glycoproteins and ECM regulators have been linked to the tumor microenvironment in various stages of gastric cancer [67,68,69,70]. Deregulation of protein expression, such as FN1 and ADAM9 (upregulated) or SPARCL-1 (downregulated), has been shown to affect cell growth and tissue proliferation in gastric cancer [70,71,72,73]. The hyperplasia seen in the gastric glands of *BmpR1a*^△FoxL1+^ mice [22] could be, in part, explained by the modification of these proteins in the microenvironment. TNC is generally absent or suppressed in most normal adult tissues, while it is markedly overexpressed in some pathological conditions, such as wound healing, inflammation, and in a variety of neoplasms [74]. This expression pattern was observed in the stomachs of *BmpR1a*^△FoxL1+^ mice when compared to that of controls. Thus, similar to gastrointestinal stromal tumors [67], whereas TNC is used as a potential marker, it can also be used as an indicator of gastric premalignancies, according to the results shown in this study.

CL is a polymeric protein present in greater quantities in the ECM under physiological conditions [75,76], as well as in the tumor microenvironment, where its extensive deposition is one of the pathological characteristics of cancers, such as gastric neoplasia [43,77]. As collagens play an important structural role in the ECM and contribute to its mechanical properties by influencing cellular behavior [78], any changes in CL organization, expression, and/or crosslinking will directly affect optimal tissue function [79]. Unexpectedly, in this study, we discovered that almost all CL chains analyzed using MS were downregulated in the *BmpR1a*^△FoxL1+^ pre-neoplastic model. This is in contrast to previous findings, especially regarding what is known from descriptive studies on ECM in gastric cancer, as well as previous studies with *BmpR1a*^△FoxL1+^ [5,22,43]. Other proteomic analyses have shown the difficulties of optimal CL protein extraction from tissues, especially when fibrotic [36,80,81]. We hypothesize that the extraction method used in this study was not optimal for CL protein analysis [81]. However, the choice of another method favoring CL protein extraction could be detrimental to the analysis of other matrisomic proteins [81]. Considering that CL chain expression, as well as its mechanical and biochemical organization, could be validated through other techniques, proteomic analyses would not be the preferred technique for studying fibrotic tissues. In this study, Sirius red staining under bright field was used for the visualization of total CL deposition in tissue, while under polarized light microscopy it provided more relevant information regarding the CL network, such as its organization, stiffness, and fiber alignment.

Altogether, the present study provides a more comprehensive representation of the evolving ECM fraction from the microenvironment in pre-neoplastic gastric lesions associated with BMP signaling-impaired TC^FoxL1+^. These findings support the importance of TC^FoxL1+^ and BMP signaling in the maintenance of a healthy microenvironment to maintain gastric homeostasis and prevent the development of pathologies such as neoplasia.

## Figures and Tables

**Figure 1 biomedicines-11-00019-f001:**
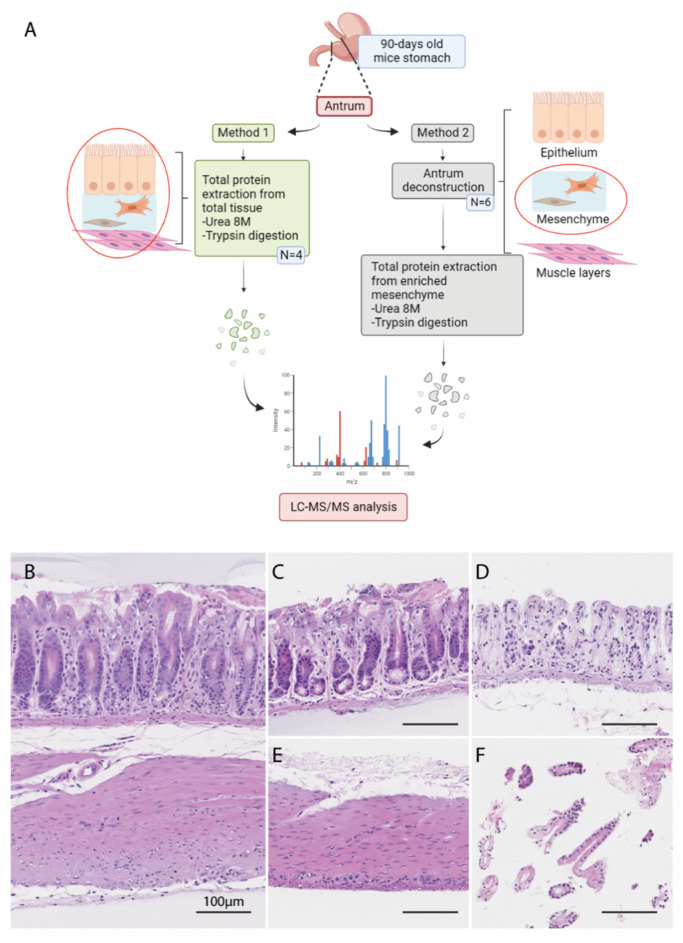
Methods for tissue preparation of stomach antrum for proteomic analysis. (**A**) Schematic representation of the experimental pipeline to assess the gastric matrisome profile in the *BmpR1a*^△FoxL1+^ mouse model. Created with BioRender.com. (**B**–**F**) Histological assessment of the deconstructed antrum tissue. Total antrum tissue (**B**) was deconstructed in a stepwise manner, where the muscle layers (**E**) were first dissociated from the other two compartments (**C**). Epithelial/mesenchymal tissue (**C**) was further dissociated, yielding the mesenchyme compartment (**D**) and the epithelium (**F**). Scale bar = 100 μm.

**Figure 2 biomedicines-11-00019-f002:**
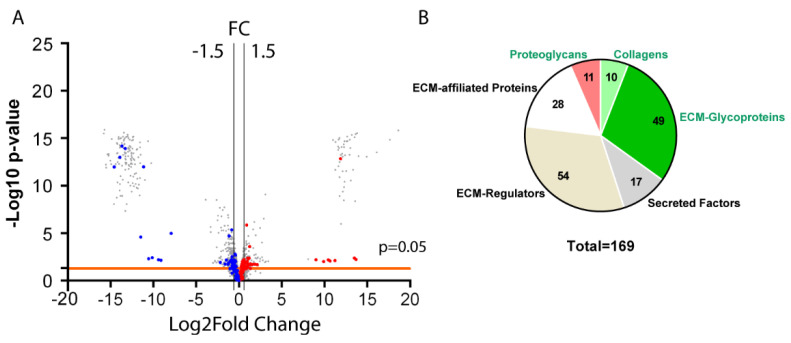
Total antrum matrisome in mice upon deletion of telocyte BMP-associated signaling. (**A**) Proteomic data from total antrum tissue isolated from control and *BmpR1a*^△FoxL1+^ mice (*n* = 4) were analyzed using ProStar to determine which proteins were significantly modulated. The volcano plot shows all differentially regulated proteins identified following mass spectrometry, highlighting significant matrisome proteins with at least a 1.5-fold change (plotted as log2FC) and a *p*-value lower than 0.05. Blue dots represent downregulated matrisome proteins; red dots represent upregulated matrisome proteins. The horizontal line represents the threshold *p*-value of 0.05. Vertical lines represent the 1.5-fold change threshold (in log2). Volcano plot was generated using GraphPad Prism version 9.4.1. (**B**). Pie chart indicates the number of matrisome proteins identified in total antrum tissue according to categories (core matrisome proteins in green and matrisome-associated proteins in black).

**Figure 3 biomedicines-11-00019-f003:**
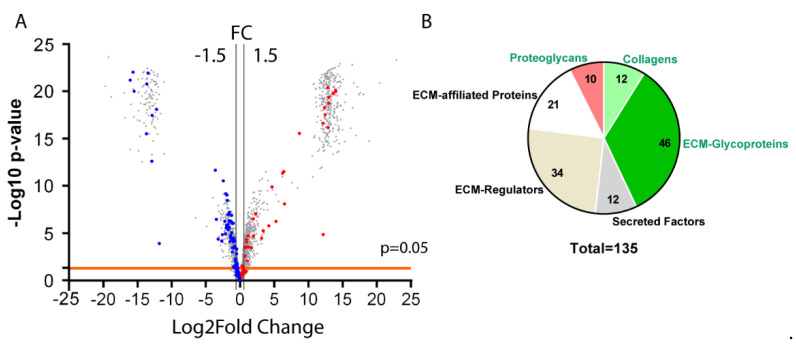
Enriched mesenchyme antrum matrisome in mice upon deletion of telocyte BMP-associated signaling. (**A**) Proteomic data from mesenchyme-enriched antrum tissue isolated from control and *BmpR1a*^△FoxL1+^ mice (*n* = 6) were analyzed using ProStar to determine which proteins were significantly modulated. The volcano plot shows all differentially regulated proteins identified following mass spectrometry, highlighting significant matrisome proteins with at least a 1.5-fold change (plotted as log2FC) and a *p*-value lower than 0.05. Blue dots represent downregulated matrisome proteins; Red dots represent upregulated matrisome proteins. The horizontal line represents the threshold *p*-value of 0.05. Vertical lines represent the 1.5-fold change threshold (in log2FC). Volcano plot was generated using GraphPad Prism version 9.4.1. (**B**) Pie chart indicates the number of matrisome proteins identified in mesenchyme-enriched antrum tissue according to categories (core matrisome proteins in green and matrisome-associated proteins in black).

**Figure 4 biomedicines-11-00019-f004:**
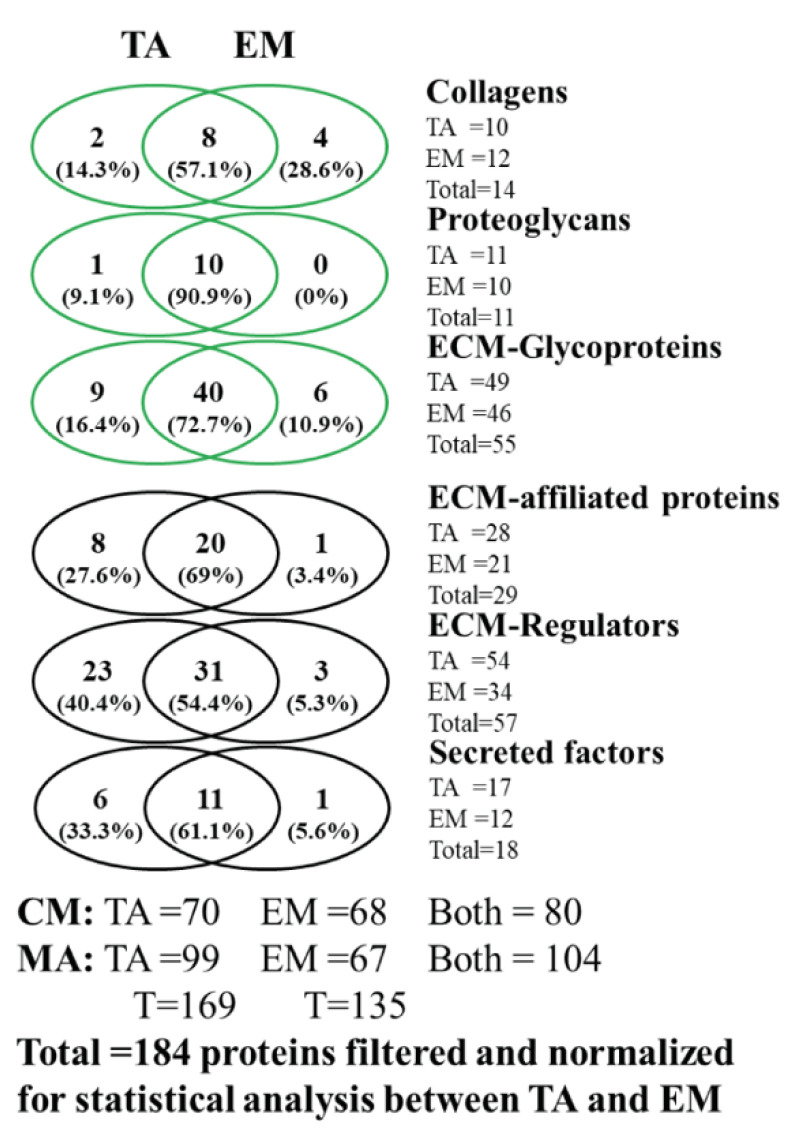
Mesenchymal enrichment of the antrum does not lead to notable ECM protein loss. Venn diagrams illustrating the overall *BmpR1a*^△FoxL1+^ mouse gastric matrisome proteins identified using the two methods combined, indicating a wide overlap between the two approaches. Core matrisome proteins are presented in green and matrisome-associated proteins are shown in black. TA, total antrum; EM, enriched mesenchyme; CM, core matrisome; MA, matrisome-associated.

**Figure 5 biomedicines-11-00019-f005:**
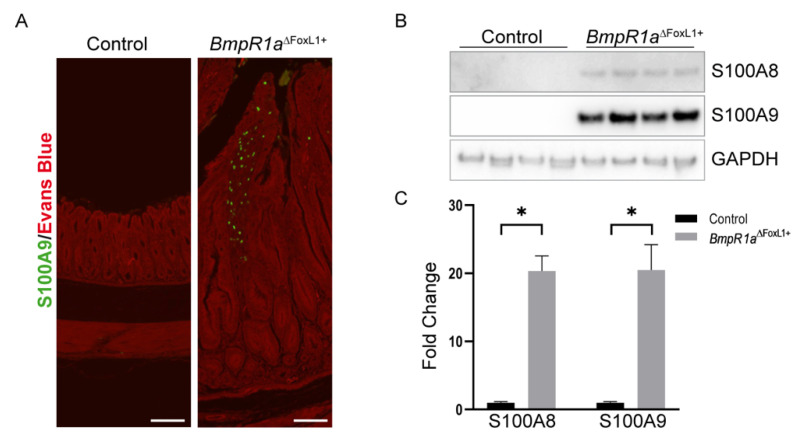
S100A8 and A9 proteins are upregulated secreted factors in *BmpR1a*^△FoxL1+^ mice, indicating an inflammatory response. (**A**) Immunostaining against S100A9 (shown in green) revealed an increase in its expression in the mesenchyme-enriched area of the antrum tissue of BmpR1a^△FoxL1+^ mice compared to that in controls. (**B**) Immunoblot analysis of the total antrum tissue indicates strong expression of both S100A8 and S100A9 proteins in BmpR1a^△FoxL1+^ mice compared to that in controls. (**C**) Quantification of immunoblots confirmed a significant increase in both S100A8 and S100A9 in the mutant animals (FC = 20.34 and 20.48, respectively) compared to that in controls. Statistical analysis was assessed using the Mann–Whitney test with * *p* < 0.05. Evans blue was used as a counterstain (red signal in (**A**)). Scale bar = 100 μm.

**Figure 6 biomedicines-11-00019-f006:**
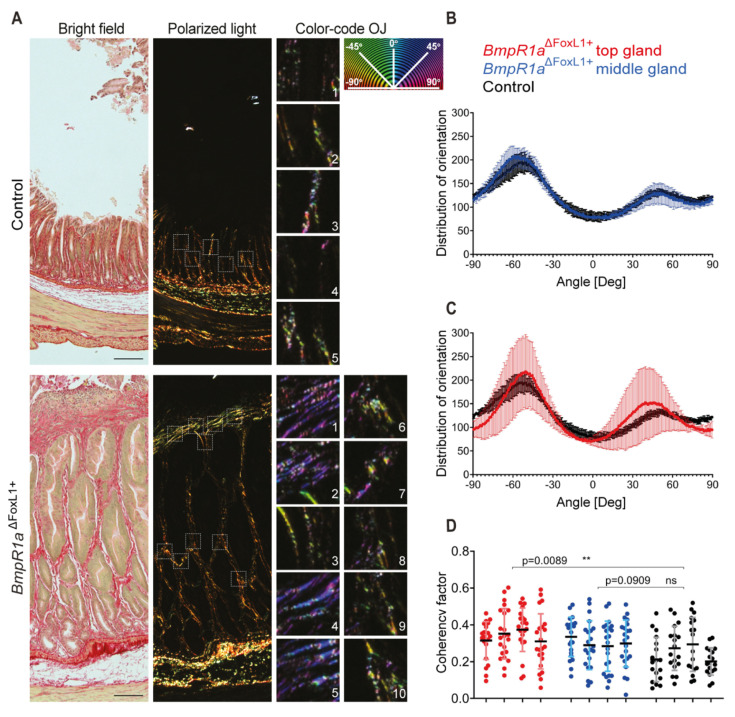
Loss of BMP signaling in gastric TC^FoxL1+^ disrupts the collagen network. (**A**) Picrosirius red staining was performed on stomach sections from both control and *BmpR1a*^△FoxL1+^ mice. Collagen fiber organization and alignment was evaluated under bright field (left panels) and polarizing light (middle panels). Imaging was performed using a Zeiss Axioscope 5 equipped with an analyzer and a linear polarizer. ROI (dotted squares) were converted to grayscale 16-bit images and color-coded where pixel hue corresponds to the angle of local fiber orientation, which ranges from −90° to +90°. Representative ROI are shown with their color-coded fiber orientation (right panels) and color-coded orientation legend is shown. (**B**) Distribution of fiber orientations was compiled for each ROI in all analyzed images, to compare control tissue with middle of the gland in *BmpR1a*^△FoxL1+^ mice antrum. Data are shown as means of distribution ± SD, for four individual mice in each group. (**C**) Distribution of fiber orientations was compiled for each ROI in all analyzed images, to compare control tissue with top of the gland in *BmpR1a*^△FoxL1+^ mice antrum. Data are shown as means of distribution ± SD, for four individual mice in each group. (**D**) Coherency factor was computed for all ROI and data were plotted showing a significant increase of fiber alignment in the top part of antrum gland in *BmpR1a*^△FoxL1+^ mice, with a mean coherency factor of 0.338 compared to 0.245 observed in control mice. No significant difference was observed between the middle part of antrum gland in control and that in *BmpR1a*^△FoxL1+^ mice. Statistical analyses were performed using Prism, with a table and group nested ANOVA. Scale bar = 50 μm. ** *p* < 0.01. ROI: representative region of interests.

**Figure 7 biomedicines-11-00019-f007:**
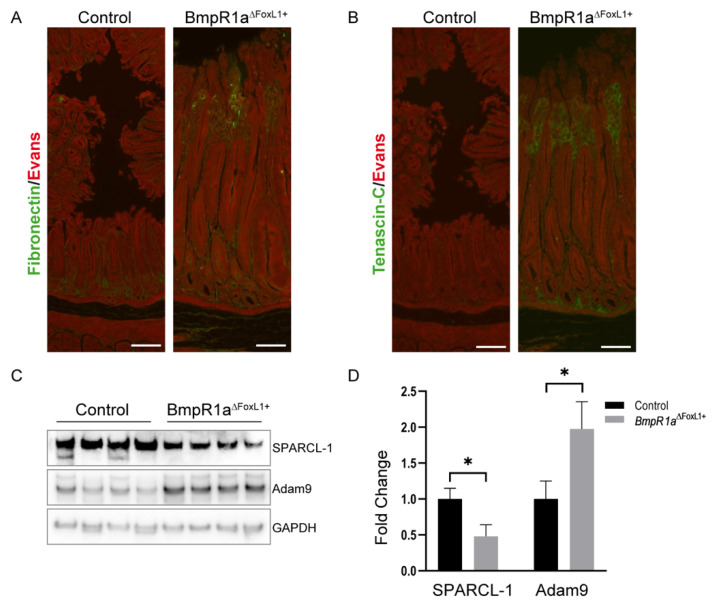
Modulations in ECM glycoproteins and ECM regulator correlate with a neoplasia phenotype in stomach of *BmpR1a*^△FoxL1+^ mice. (**A**). Immunostaining against ECM glycoprotein fibronectin (shown in green) revealed an increased expression in the enlarged mesenchymal area of the antrum tissue in *BmpR1a*^△FoxL1+^ mice compared to that in controls. (**B**) Immunostaining against ECM glycoprotein Tenascin C (shown in green) revealed an increased expression in the antrum mesenchyme of *BmpR1a*^△FoxL1+^ mice compared to that in controls. (**C**). Immunoblot analysis showed a decrease of the ECM glycoprotein SPARCL-1 expression and an increase of the ECM regulator ADAM9 in total antrum samples of *BmpR1a*^△FoxL1+^ mice compared to that in controls. GAPDH was used as a loading control. (**D**) Quantification of immunoblots revealed a significant modulation of SPARCL-1 and ADAM9 between both group (FC = 0.48 and 1.98, respectively). All quantifications were performed using ImageJ and statistical analyses were performed using Prism. All immunoblot quantification data are presented as the mean ± SD (*n* = 4). Statistical analysis was assessed using the Mann–Whitney test with * *p* < 0.05. Evans blue was used as a counterstain (red signal in A and B). Scale bar = 100 μm.

**Table 1 biomedicines-11-00019-t001:** Total antrum tissue.

Core Matrisome						Matrisome-Associated					
ECM-Glycoproteins			Collagen Chains			ECM-Regulators			ECM-Affiliated Proteins		
Name	FC	*p*-Value	Name	FC	*p*-Value	Name	FC	*p*-Value	Name	FC	*p*-Value
Agrn	2372	0.0076	Col18a1	1.94	0.0099	Adam9	510	0.0063	Muc4	1594	0.0084
Dmbt1	3.60	0.0188	Col15a1	1.19	NC/0.1689	Prss3	2.51	NC/0.0528	Mbl2	1382	0.0066
Fgb	2.96	0.0200	Col6a5	−1.15	0.4277	Leprel1	2.38	NC/0.0002	Lgals7	4.40	NC/0.0210
Fgg	2.67	0.0206	Col4a1	−1.36	0.1037	Serpinb1a	2.29	0.0044	Lgals4	2.09	0.0276
Fga	2.54	0.0175	Col12a1	−1.73	0.0030	Serpinb5	2.07	0.0183	Lgals9	1.63	0.0293
Vtn	2.49	0.0191	Col14a1	−2.08	0.0558	Ctss	2.05	0.0043	Anxa3	1.54	0.0224
Thbs1	2.10	0.0049	Col6a1	−3.10	0.0178	Ctsc	1.85	0.0134	Anxa10	1.51	0.0454
Mfge8	1.51	0.071	Col6a2	−4.52	0.0124	Serpinb12	1.84	NC/0.0554	Anxa1	1.48	0.0121
Tnc	1.40	0.0253	Col4a2	−553	0.0071	Ctsh	1.73	0.0113	Lgalsl	1.43	0.0206
Igfbp7	1.39	0.1144	Col1a2	−1496	0.0049	F2	1.56	0.0280	Reg1	1.35	0.6373
Creld2	1.35	0.1078				Try10	1.53	NC/0.0394	Hpx	1.31	0.0656
Fn1	1.33	0.0094	**Proteoglycans**			Hrg	1.51	0.0357	Reg2	1.29	NC/0.1715
Vwa1	1.28	NC/0.0437	**Name**	**FC**	***p*-Value**	Plg	1.39	0.0466	Anxa2	1.26	0.0574
Fbln2	1.24	0.174	Bgn	1.31	0.0160	Ctse	1.38	0.0548	Anxa7	1.25	0.0696
Sparc	1.23	0.1481	Prg2	1.09	0.7556	Serpinf2	1.37	0.0572	Lman1	1.12	0.4453
Postn	1.22	0.0344	Vcan	−1.27	0.0841	Itih3	1.36	0.0472	Anxa5	1.10	0.1127
Lrg1	1.14	0.2343	Hspg2	−1.39	0.0302	Ctsa	1.35	0.0365	Anxa4	1.08	0.2235
Vwa5a	1.13	0.1869	Prelp	−1.41	0.0513	Serpini2	1.34	NC/0.3921	Muc6	1.05	0.7641
Tgfbi	1.12	0.1154	Lum	−1.47	0.0188	Serping1	1.33	0.0997	Sema4b	1.01	0.9016
Aebp1	1.03	0.8022	Aspn	−1.56	0.0104	Serpinc1	1.30	0.1275	Lgals3	−1.03	0.7888
Efemp1	1.02	0.7931	Ogn	−1.85	0.0065	Itih2	1.26	0.1419	Anxa11	−1.05	0.3729
Ltbp4	1.01	0.906	Dcn	−2.05	0.0094	Kng1	1.24	0.0814	Plxnb2	−1.06	0.3421
Fbln1	−1.01	0.8928	Podn	−2835	3 × 10^−5^	F13a1	1.23	0.0278	Lgals1	−1.10	0.2588
Thbs4	−1.12	NC/0.5325	Fmod	−12,933	7 × 10^−15^	Cst3	1.23	0.0257	Anxa6	−1.47	0.0357
Nid2	−1.20	0.1529				Adam10	1.21	0.1819	Muc5ac	−1.50	0.0477
Fbln5	−1.23	0.1405				Ctsb	1.17	0.1297	Sdc1	−1.55	NC/0.0175
Lamb1	−1.27	0.0300				Ctsl	1.15	0.1851	Lgals2	−2.76	0.0067
Pcolce	−1.31	NC/0.0097				Ctsz	1.14	0.1260	Cspg4	−9977	1 × 10^−14^
Lamb3	−1.32	NC/0.0020				A2m	1.13	0.3687			
Sbspon	−1.34	NC/0.0423				Itih1	1.13	0.2482	**Secreted factors**		
Tsku	−1.44	NC/0.0188				Serpinb9	1.10	0.4599	**Name**	**FC**	***p*-Value**
Dpt	−1.58	0.0069				Serpina1e	1.08	0.8936	S100a9	13058	0.0059
Mfap4	−1.60	0.0210				Serpinh1	1.06	0.5231	S100a8	11412	0.0043
Adipoq	−1.61	0.0623				Ctsd	1.05	0.4297	Sfrp1	3720	NC/1 × 10^−13^
Lama4	−1.67	0.0145				Cstb	1.04	0.5912	Il1rn	948	0.0100
Lamc1	−1.69	0.0269				Ngly1	1.03	0.8976	S100a6	1.99	0.0389
Mfap5	−1.69	0.0429				Serpinf1	−1.00	0.9864	Rptn	1.86	NC/1 × 10^−13^
Nid1	−1.70	0.0116				Serpind1	−1.02	0.8891	S100g	1.58	0.0410
Lama2	−1.70	0.0377				Cela2a	−1.05	0.9419	S100a4	1.52	0.0059
Tinagl1	−1.74	0.0283				Fam20b	−1.06	0.5655	S100a1	1.46	0.0119
Lama5	−1.77	0.0121				St14	−1.09	0.4949	S100a13	1.38	0.0825
Emilin1	−1.79	NC/4 × 10^−6^				Cela3b	−1.11	0.8716	S100a11	1.36	0.0774
Lamb2	−2.33	0.0140				Serpina3k	−1.18	0.5193	S100a14	1.32	0.0266
Tnxb	−2.38	0.0193				Prss2	−1.20	0.7794	Il18	1.20	0.2779
Fbn1	−243	1 × 10^−5^				Serpina1d	−1.25	0.1443	S100a16	1.18	0.1155
Abi3bp	−1117	0.0038				Tgm2	−1.29	0.0344	Hcfc1	−1.12	0.0848
Mfap2	−2262	NC/1 × 10^−12^				Serpina1c	−1.30	0.3113	S100a10	−1.23	0.0344
Sparcl1	−15,377	1 × 10^−13^				Cela1	−1.30	0.6149	S100b	−2.20	1 × 10^−5^
Spp1	−24,277	NC/1 × 10^−12^				F12	−1.38	0.0016			
						P4ha1	−1.41	NC/0.0071			
						Serpina1b	−1.47	0.1053			
						P4ha2	−1.63	NC/0.0159			
						Serpina6	−1.76	0.1680			
						Ambp	−671	0.0064			

**Table 2 biomedicines-11-00019-t002:** Total Enriched mesenchyme from antrum tissue.

Core Matrisome						Matrisome-Associated					
ECM-Glyucoproteins			Collagen Chains			ECM-Regulators			ECM-Affiliated Proteins		
Name	FC	*p*-Value	Name	FC	*p*-Value	Name	FC	*p*-Value	Name	FC	*p*-Value
Fbln1	5535	3 × 10^−18^	Col18a1	2.28	2 × 10^−5^	Serpinb5	15,966	7 × 10^−21^	Muc4	7357	4 × 10^−21^
Dmbt1	74,2	5 × 10^−12^	Col4a1	−1.17	0.4016	Plg	8304	4 × 10^−20^	Muc5ac	409	NC/3 × 10^−16^
Fgb	18,4	2 × 10^−6^	Col6a4	−1.49	0.0575	Mmp9	8201	2 × 10^−19^	Anxa10	90.47	8 × 10^−9^
Fgg	10,4	6 × 10^−6^	Col4a2	−1.72	0.0288	Loxl2	7570	NC/6 × 10^−17^	Lgals4	25.55	1 × 10^−10^
Fga	8.91	3 × 10^−5^	Col15a1	−2.72	2 × 10^−5^	Fam20b	5265	5 × 10^−19^	Lgals9	4.89	9 × 10^−8^
Tnc	1.96	9 × 10^−5^	Col6a2	−2.84	6 × 10^−7^	Adam10	4586	2 × 10^−5^	Lgals2	3.87	2 × 10^−5^
Vtn	1.93	NC/4 × 10^−5^	Col6a1	−2.96	3 × 10^−7^	P4ha2	4433	NC/2 × 10^−17^	Anxa3	1.74	0.0003
Mfge8	1.78	0.0003	Col6a3	−3.08	1 × 10^−7^	Ctse	3.81	3 × 10^−7^	Anxa11	1.35	0.0461
Fn1	1.46	0.0760	Col1a2	−6.44	7 × 10^−5^	Cst3	2.08	NC/0.0088	Anxa7	1.16	0.1634
Fbln5	1.41	NC/0.0454	Col1a1	−9.15	4 × 10^−5^	Serpinc1	1.88	0.1149	Lman1	1.15	0.2975
Igfbp7	1.30	0.1291	Col6a5	−10.97	3 × 10^−7^	F13a1	1.66	0.1621	Plxnb2	1.12	0.4665
Ecm1	1.10	NC/0.3423	Col4a6	−12,771	NC/2 × 10^−21^	Serpinb9	1.64	NC/0.0025	Anxa4	−1.07	0.4975
Creld2	1.09	0.6556				Serpinb1a	1.27	0.0296	Anxa1	−1.17	0.2537
Ltbp4	−1.04	0.8548	**Proteoglycans**			A2m	1.20	0.1905	Muc6	−1.22	0.2380
Tgfbi	−1.06	0.5927	**Name**	**FC**	***p*-Value**	Ctsc	−1.05	0.6744	Cspg4	−1.25	NC/0.0437
Agrn	−1.08	0.4679	Prg2	2.27	0.0003	Itih1	−1.08	0.7179	Lgals3	−1.28	0.0832
Vwf	−1.32	0.0853	Hspg2	−1.4	0.0134	Ctsh	−1.12	0.2113	Anxa2	−1.31	0.0187
Vwa1	−1.32	NC/0.0617	Bgn	−1.50	0.1212	Ctsb	−1.13	0.2275	Sema3d	−1.44	NC/0.0015
Postn	−1.58	0.0054	Podn	−2.23	0.0010	Ctsa	−1.20	0.1233	Anxa5	−2.05	5 × 10^−5^
Vwa5a	−1.61	0.0076	Prelp	−3.01	5 × 10^−7^	Serpina1c	−1.22	0.1968	Lgals1	−2.78	8 × 10^−5^
Mfap4	−1.61	0.0311	Aspn	−3.42	1 × 10^−7^	Ctsz	−1.26	0.0404	Anxa6	−3.53	2 × 10^−6^
Lamb1	−1.73	0.0002	Dcn	−3.62	3 × 10^−9^	Itih3	−1.30	0.2241			
Emilin1	−1.79	0.0432	Lum	−3.78	9 × 10^−10^	Ctsd	−1.40	0.0005	**Secreted factors**		
Adipoq	−1.95	NC/0.0357	Ogn	−4.29	7 × 10^−10^	Itih2	−1.49	0.2789	**Name**	**FC**	***p*-Value**
Papln	−1.96	NC/9 × 10^−7^	Vcan	−12.30	2 × 10^−12^	Cstb	−1.52	0.0248	S100a16	16,177	1 × 10^−20^
Aebp1	−2.00	0.0004				Serpinh1	−1.66	0.0003	S100a14	12,545	2 × 10^−20^
Nid2	−2.09	9 × 10^−7^				Serpina3k	−1.68	NC/7 × 10^−5^	S100a9	85.2	NC/3 × 10^−12^
Lamc1	−2.33	9 × 10^−6^				Serping1	−1.93	NC/3 × 10^−5^	S100a8	37.9	6 × 10^−7^
Lama4	−2.34	1 × 10^−7^				P4ha1	−2.01	8 × 10^−5^	S100a1	3.1	NC/0.0003
Nid1	−2.50	5 × 10^−7^				Ctss	−2.13	5 × 10^−5^	S100a4	1.3	0.2283
Sbspon	−2.54	NC/3 × 10^−5^				Tgm2	−2.88	6 × 10^−8^	Angptl2	1.22	NC/0.0289
Lama5	−2.57	2 × 10^−5^				Cela1	−3.38	4 × 10^−6^	Hcfc1	1.20	0.2782
Tinagl1	−3.45	1 × 10^−6^				Ambp	−13,221	3 × 10^−16^	S100a6	1.20	0.1979
Lamb2	−3.98	1 × 10^−6^				Adamts20	−68,487	NC/6 × 10^−22^	S100a11	1.12	0.3482
Mfap5	−4.09	3 × 10^−6^							S100a13	−1.12	0.4822
Tnxb	−4.37	1 × 10^−5^							S100a10	−2.16	0.0004
Lama2	−4.66	6 × 10^−7^									
Dpt	−5.49	3 × 10^−11^									
Fbn1	−5.99	2 × 10^−5^									
Sparc	−3620	0.0001									
Spp1	−4710	NC/8 × 10^−19^									
Mmrn2	−7450	3 × 10^−18^									
Mfap2	−7588	3 × 10^−13^									
Abi3bp	−11,200	1 × 10^−22^									
Fbn2	−46,287	1 × 10^−20^									
Spon1	−51,711	NC/9 × 10^−23^									

## Data Availability

Raw files, databases, and MaxQuant results have been deposited in ProteomeXchange with the accession number PXD038603.

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
