# Peer review of "Comprehensive Profiling of Early Neoplastic Gastric Microenvironment Modifications and Biodynamics in Impaired BMP-Signaling FoxL1+-Telocytes"

_biomedicines, 2022, doi:10.3390/biomedicines11010019_

Round 1

Reviewer 1 Report

The manuscript entitled “Comprehensive Profiling of early Neoplastic Gastric Microenvironment Modifications and Biodynamics in Impaired BMP Signaling Foxl1+-Telocytes” provides a better understanding of how TCFoxL1+ defective BMP signaling participates in gastric pre-neoplastic microenvironment, using a proteomic approach to determine the changes in the complete matrisome of BmpR1aâ–³FoxL1+ and control mice, both in total antrum as well as in isolated mesenchyme-enriched antrum fractions. These results show that TCFoxL1+ are key members of the mesenchymal cell population and actively participate in the establishment of the matrisomic fraction of the microenvironment; thus, influencing epithelial cell behavior.

 Using this model, it can perform a matrisomic investigative of the stomach of control and BmpR1aâ–³FoxL1+ mice, and better understand the contribution of TCFoxL1+ to this aspect of the microenvironment.

English language and style need to be well checked.

Author Response

Reviewer 1

We thank Reviewer 1 for the provided comment. The manuscript has been reviewed for its English language and style.

Reviewer 2 Report

This is a neat study on the proteomic profiling of the gastric cancer ECM following impaired BMP signaling in Foxl1+-Telocytes. The study is carefully conducted and the data generally support its conclusion. Nevertheless, some additional experiments could be performed as follows:

1. The authors identified S100A8/9 as ECM markers. However, these genes are predominant factors secreted by immune cells. Does the authors have any idea on the immune cells, e.g. macropohages, MDSCs, contributing to S100A8/9 secretion in their model? Please performed additional IF staining of key immune cells. 

2. One of the limitation of the study is that all analysis are essential descriptive whilst lacking in mechanistic details. It would be a great addition if any validation work can be performed, e.g. ADAM9, that could potentially contribute to GC development.

Author Response

Reviewer 2

We thank Reviewer 2 for the provided comments. All specific comments to the authors were taken into consideration.

  1. The authors identified S100A8/9 as ECM markers. However, these genes are predominant factors secreted by immune cells. Does the authors have any idea on the immune cells, e.g. macropohages, MDSCs, contributing to S100A8/9 secretion in their model? Please performed additional IF staining of key immune cells.

Following the reviewers comment we noticed that we did not commented the reservoir function of the ECM and how it impact secreted factors. Factors such as S100A8/9 were studied as part of this component of the matrisome that is included in the in silico evaluation. In fact, S100A8/9 are known in this context as secreted factors associated to the ECM. This specification is now mentioned in the text and the importance of the reservoir function is introduce at line 459.

  1. One of the limitation of the study is that all analysis are essential descriptive whilst lacking in mechanistic details. It would be a great addition if any validation work can be performed, e.g. ADAM9, that could potentially contribute to GC development.

Validation of ADAM9 is provided by WB in figure 7. Also deregulated in gastric cancer and validated in this study in figure 7 are SPARCL-1 (WB) FN and TNC (both validated by IF). Precision on gastric cancer is now provided in the discussion line 670-672. An additional reference of FN is now provided (ref. 73).